# Effect of Zilpaterol Hydrochloride and Zinc Methionine on Growth, Carcass Traits, Meat Quality, Fatty Acid Profile and Gene Expression in *Longissimus dorsi* Muscle of Sheep in Intensive Fattening

Manuel Guerrero-Bárcena [1], Ignacio Arturo Domínguez-Vara [1,*], Ernesto Morales-Almaraz [1], Juan Edrei Sánchez-Torres [1], José Luis Bórquez-Gastelum [1], Daniel Hernández-Ramírez [1], Daniel Trujillo-Gutiérrez [1], Miguel Angel. Rodríguez-Gaxiola [2], Juan Manuel Pinos-Rodríguez [3], Gisela Velázquez-Garduño [4] and Fernando Grageola-Nuñez [5]

[1] Departamento de Nutrición Animal, Facultad de Medicina Veterinaria y Zootecnia, Universidad Autónoma del Estado de México, Campus Universitario "El Cerrillo", Toluca 50090, CP, Mexico; fosforo620@hotmail.com (M.G.-B.); maernesto@hotmail.com (E.M.-A.); edreie@yahoo.com.mx (J.E.S.-T.); jlborquez@yahoo.com.mx (J.L.B.-G.); qfbdaniel@yahoo.com (D.H.-R.); danieltg_dan@yahoo.es (D.T.-G.)

[2] Facultad de Medicina Veterinaria y Zootecnia, Universidad Autónoma de Sinaloa, Gral. Ángel Flores Pte. S/N Col. Centro, Culiacán 80000, CP, Mexico; m_angel2412@hotmail.com

[3] Facultad de Medicina Veterinaria y Zootecnia, Universidad Veracruzana, Xalapa 91710, VC, Mexico; jpinos70@hotmail.com

[4] Unidad Académica Capulhuac, Universidad Tecnológica del Valle de Toluca, Toluca 52700, CP, Mexico; gisela.velazquez@utvtol.edu.mx

[5] Unidad Académica de Medicina Veterinaria y Zootecnia, Universidad Autónoma de Nayarit, Ciudad de la Cultura "Amado Nervo", Compostela 63155, CP, Mexico; fggrageola7@hotmail.com

* Correspondence: igy92@hotmail.com; Tel.: +52-722-296-5542; Fax: +52-722-296-5548

**Abstract:** Zilpaterol hydrochloride (ZH) redistributes ingested energy and improves feed efficiency by increasing muscle mass and reducing fat in sheep and cattle carcasses in fattening; however, by increasing lipolysis and reducing intramuscular fat (IMF), it can affect meat quality in terms of the attributes of tenderness, juiciness, taste and color; in contrast, Zn methionine (ZM), due to its lipogenic effect, can improve meat marbling without affecting production efficiency. In the current study, 36 male Suffolk sheep were used ($25 \pm 0.58$ kg live weight, LW) to evaluate the supply of ZH and ZM on growth, carcass traits, meat quality, fatty acid content and expression of genes which regulate the deposition of fatty acids (FA) in IMF. A completely randomized design was used, with factorial arrangement of $2 \times 2$ ZH (0 and 0.2 mg kg$^{-1}$ LW) and ZM (0 and 80 mg Zn kg$^{-1}$ dry matter, DM). The results showed that ZH increased ($p < 0.05$) carcass yield, compactness index and chop eye area and decreased greasing ($p < 0.02$). The content of ether extract in meat increased ($p < 0.05$) in sheep with ZM plus ZH, and in sheep with ZM ($p < 0.01$). ZH ($p < 0.05$) reduced ($p < 0.02$) the meat's color index L*, a*, b*, C* and H*. The content in IMF of stearic (C18:0) and arachidic (C20:0) FA was reduced ($p \leq 0.05$) by the effect of ZH, but the palmitoleic (C16:1), eicosatetraenoic (C20:4n6) and conjugated linoleic FA were increased ($p \leq 0.05$) by the effect of ZH. ZM increased ($p \leq 0.05$) palmitoleic (C16:1) and conjugated linoleic FA; the ZH interaction with ZM increased ($p \leq 0.05$) linoleic (C18:2 c 9 c 12), linolenic (C18:3 c 9c12c15) and eicosatetraenoic (C20:4n6) FA. The ZH interaction with ZM influenced ($p \leq 0.05$) the total saturated fatty acids (SFA), unsaturated fatty acids (UFA) and polyunsaturated fatty acids (PFA). ZH increased ($p \leq 0.05$) the relative expression of mRNA from the enzymes lipoprotein lipase (LPL), hormone-sensitive lipase (HSL), glycerol-3-phosphate acyltransferase (GPAT1) and diglyceride acyltransferase (DGAT1). ZM increased ($p \leq 0.05$) the relative expression of mRNA from the enzyme gene acetyl-CoA carboxylase (ACC) and HSL, monoglyceride lipase (MGL). The ZM interaction with ZH increased ($p \leq 0.05$) the relative expression of mRNA genes of the enzymes HSL and ACC. It was concluded that ZH improved feed conversion (FC), increased yield and reduced fat in carcasses; ZM increased IMF in *Longissimus dorsi*. ZH and ZM influenced the FA composition, reduced the SFA and increased the UFA and PFA; both additives also influenced the relative mRNA expression of genes involved in fatty acid metabolism.

**Keywords:** sheep; zilpaterol; zinc; meat; fatty acids; lipogenic and lipolytic genes

## 1. Introduction

Carcass evaluation is essential to assigning values in meat quality; subcutaneous fat and chop eye area are indicators of meat quality and marketable carcass quantity [1], and IMF and marbling degree are indicators of high-quality standards to satisfy the consumer [2]. The minimum IMF content for good palatability is 3% wed base (WB) [3]; in sheep, the highest IMF deposition is at an early age and at sexual maturity [4,5]; in general, it is less than 5% [6], but at an advanced age, they deposit a lot of subcutaneous fat. Therefore, the meat industry requires alternatives to increase cattle and sheep IMF.

The β-adrenergic agonists (βAAs) redistribute dietary energy by promoting its storage in specific sites of the carcass; βAAs are chemical agents that act at the level of adrenergic receptors which are associated with G protein, deriving, in cellular metabolism, energy from food and lipolysis towards protein synthesis in striated muscle [7,8]. ZH is a βAA authorized for commercial use in livestock in Mexico, South Africa, the United States (FDA, NDA, 2006, 141–258) and Canada [9,10]. In fattening cattle, ZH has positive effects on productive response and carcass yield [11,12]; in fattening sheep, ZH increases carcass yield, *Longissimus dorsi* (LD) muscle area, increases muscle proportion and reduces fat and bone proportion in the carcass; it can also influence meat coloration indices [13]. Regarding lipid content, Choi et al., Van Bibber-Krueger, and Rodríguez [14–16] indicated that FA content in fattening cattle's LD muscle was not affected by ZH. In fattening sheep, Dávila-Ramos and Robles-Estrada [17] indicated that ZH reduced the content of saturated FA and increased the content of the unsaturated FA in the LD muscle, and reduced as lower proportion of stearic acid when supplementing ZH. Therefore, βAAs, by stimulating lipolysis and reducing IMF, can affect meat quality attributes such as color, juiciness and tenderness [18,19].

Zinc (Zn) is required for the action of more than 300 metalloenzymes [20], several related to lipid metabolism [21]. IMF deposition is mainly regulated by lipid metabolism, thus determining the balance between its storage and removal in the LD muscle. Therefore, the expression of lipid metabolism genes includes those enzymes involved in lipogenic processes [acetyl-CoA carboxylase (ACC) and FA synthase (FASN)], FA uptake [lipoprotein lipase (LPL)], FA esterification [lipoprotein lipase (LPL), glycerol-3-phosphate acyltransferase (GPAT1) and diglyceride acyltransferase (DGAT1)] and FA lipolysis [hormone-sensitive lipase (HSL) and monoglyceride lipase (MGL)] [22], which influence the IMF and marbling of beef [23], and the IMF, marbling, tenderness and FA content of sheep meat [24].

The bioavailability of trace minerals depends on the source, levels of absorption, retention and utilization of the animal; organic sources, in theory, are more similar to the forms in which minerals are in the body; therefore, they are more stable in the digestive tract and more bioavailable than inorganic ones; organic minerals, being protected, react less and do not form insoluble complexes with other compounds in a diet that reduces their absorption [25]. Organic minerals or metal complexes are the product of the union of a metal ion to a ligand: a molecule or ion with an atom with a pair of free electrons. According to the Association of American Feed Control Officials [26], Zn-methionine in a specific metal amino acid complex (the product of the reaction of a soluble metal salt and a specific amino acid), it contains a very consistent specific chemical structure [27].

However, it is not known whether ZH and ZM influence the expression of these lipid metabolism genes.

Zn is involved in carbohydrate metabolism, is necessary for glycolysis and activates the enzyme pyruvate kinase, which catalyzes the reaction of phosphoenolpyruvate to pyruvate [28]; insulin secretion [21,29], proinsulin, assembled in the Golgi apparatus, requires two Zn ions to be secreted [30]. Zn has lipogenic effects [22,31] and can simulate some actions of insulin and increase glucose use, thus causing increased lipogenesis [32]. In obese mice, Zn increases body fat and reduces feed intake [32]; in cattle, Spears and Kegley [23]

indicated the effects of organic Zn from zinc proteinate on weight gain, carcass yield and marbling degree. Rodríguez-Gaxiola et al. [12] reported that including Zn methionine did not affect the back fat, fat in kidney and fat in pelvic cavity. In contrast, Malcolm-Callis et al. [22] indicated that 30 mg of Zn $kg^{-1}$ DM of the complex Zn amino acids in diets for steers had a more significant effect on cover fat and cavity fat than inorganic Zn sources. However, Greene et al. [33] observed steers' Zn methionine (82 mg Zn $kg^{-1}$ DM) promoted greater external fat and carcass quality. Likewise, Spears and Kegley [23] indicated that, in finishing steers, 83 mg $kg^{-1}$ DM of Zn proteinate increased hot carcass weight and chop area. Regarding the effect of Zn-Met on meat color, Rodríguez-Gaxiola [12] observed that Zn-Met reduced ($p$ = 0.06) the intensity of the parameter $b^*$, which may be due to the effect of Zn on the oxidation of myoglobin to metmyoglobin [34].

With respect to Zn requirements, the NRC [35] recommends 24 to 51 mg Zn $kg^{-1}$ DM in growing and finishing sheep (20 to 40 kg LW); NAS [36] recommends 30 mg Zn $kg^{-1}$ DM in fattening cattle; however, studies have shown that, for greater lipogenesis, IMF and marbling degree in the meat of sheep and cattle in intensive fattening with diets high in grains, the dietary dose of Zn may be higher than the suggested values. According to Rodríguez-Maya et al. [24], 65 mg of Zn $kg^{-1}$ DM from Zn methionine in sheep diet improved weight gain and feed conversion, increased leg circumference, IMF and marbling degree in the LD muscle and reduced lipid oxidation in cooked meat. In the same study, Zn influenced the content of myristic, palmitoleic and arachidonic FA, but did not affect the total SFA and UFA of the meat. In cattle, 80 mg Zn $kg^{-1}$ DM increased back fat and IMF, but when adding ZH, the values decreased considerably, possibly due to the lipolytic effect of ZH [12]. Reduced adipose tissue results in increased disposition of dietary lipids in non-adipose tissues such as muscle and liver [37].

Therefore, the objective of this study was to evaluate the effect of ZH and ZM supplied in the diet on productive efficiency, carcass traits, meat quality, fatty acid content and expression of genes related to lipid metabolism in the *Longissimus dorsi* of sheep in an intensive fattening system.

## 2. Materials and Methods

The study was carried out in the Faculty of Veterinary Medicine and Zootechnics of the Autonomous University of the State of Mexico, University Campus "El Cerrillo", Toluca, Mexico.

### 2.1. Animals, Treatments and Experimental Conditions

Thirty-six Suffolk sheep were used with an average live weight (LW) of 25 ± 0.58 kg and four months of age, housed in individual pens (1.5 × 2.5 m), equipped with a feeder and automatic drinker, which were fed twice a day (8:00 and 15:00 h) with a basal diet (BD, g $kg^{-1}$ DM) according to the NRC [35] (Table 1), during 89 d (adaptation 1–14 d and measurement 15–89 d) of fattening. Before the start of the experimental phase, the initial weight (kg) was recorded, and each sheep was dewormed, immunized against clostridia and administered intramuscularly vitamin B complex and vitamins A, D and E. The total sheep were divided into four groups with 9 repeats, randomly assigned: (1) Control (BD), (2) Zinc-methionine (BD + ZM, 65 mg of Zn $kg^{-1}$ DM), (3) Hydrochloride zilpaterol (BD + ZH, 0.2 mg ZH $kg^{-1}$ PV $d^{-1}$) and (4) BD + ZM, 65 mg of Zn $kg^{-1}$ DM + ZH, 0.2 mg CZ $kg^{-1}$ LW $d^{-1}$. Sheep were supplemented throughout the measurement period with Zn (Availa Zn 120, Zinpro Corporation; Eden Prairie, MN, USA), and from 42 to 71 d with (Zilmax®, Schering-Plough Animal Health, Summit, NJ, USA). Daily doses of ZH and ZM from each sheep were provided at the feeder, mixing the product with the top of the fresh offered morning feed. During the experiment, dry matter intake (DMI) of each sheep was recorded; half of the ration was offered at 0800 and the rest at 1500 h, with free access to clean and fresh water. The sheep were weighed every two weeks; the initial LW was recorded on day 15 of the test, and the final LW on day 89, before slaughter. In addition, daily weight gain (DWG) and feed conversion (FC) were

calculated. Throughout, the sheep were managed following the Federal Law on Animal Health and NOM-062-Z00-1999 [38]. Samples of the supplied feed were collected (Table 1) and the crude protein content (method 107976.05) [39], Ca, P, Zn (Perking Elmer, 3110) [40] and NDF [41] modified for use in Ankom fiber analyzer equipment (Ankom Technology, Fairport, NY, USA) were analyzed.

**Table 1.** Composition of the basal diet supplied and calculated chemical analysis.

| Composition | (g kg$^{-1}$ DM) |
|---|---|
| Milled corn | 600.00 |
| Corn stover | 160.00 |
| Soybean meal | 140.00 |
| Wheat bran | 60.00 |
| Sodium bicarbonate | 15.00 |
| Pre-mixed vitamins and minerals [1] | 25.00 |
| Chemical composition | |
| Dry matter (g kg$^{-1}$) [2] | 876.80 |
| Crude protein (g kg$^{-1}$ DM) [2] | 140.80 |
| Neutral detergent fiber (g kg$^{-1}$ DM) [2] | 214.60 |
| Metabolizable energy (MJ kg$^{-1}$ DM) [3] | 11.83 |
| Net energy for gain (MJ kg$^{-1}$ DM) [3] | 5.32 |
| Calcium (g kg$^{-1}$ DM) [4] | 7.50 |
| Phosphorus (g kg$^{-1}$ DM) [5] | 3.62 |
| Zinc (mg kg$^{-1}$ DM) [4] | 18.03 |

[1] Pre-mixed: Macrominerals (g) Ca 4500; Mg 36; K 90; Na 125. Microminerals (g) Cu 20; Fe 140; Co 500; I 500 mg; Se 90 mg. Vitamins (UI kg$^{-1}$): Vitamin A 3000; Vitamin D$_3$ 750; Vitamin E 5. [2] Determined in the laboratory [39,41]. [3] Calculated from the composition of dietary ingredients [35]. [4] Determined in the laboratory by flame atomic absorption spectrophotometry (Perkin Elmer, 3110). [5] Determined by colorimetry [40].

## 2.2. Carcass Traits

At the end of the fattening period, the sheep were fasted (16 h) with free access to water available and were transported (distance of 54 km, 1.5 h) to a slaughterhouse, with 4 h of rest before slaughter, complying with the provisions of the standards of the Official Mexican Standard (NOM-033-SAG/ZOO-2014) [42]. The sheep were weighed before slaughter (LWS) and after slaughter the carcasses were weighed to obtain the hot carcass weight (HCW); carcass yield (CY) was obtained with LWS and HCW. The carcasses were cooled at 4 °C for 24 h to record the weight of the cold carcass (CCW) and then the morphometry (linear measurements) was evaluated with a tape measure and a vernier. Subcutaneous fat, renal fat and carcass muscle conformation variables were evaluated according to the standards of the European Community [43] and the Official Mexican Standard (NMX-FF-106 SCFI-2006) [44]. Subsequently, a cross-section was made in the LD muscle, on the left side of the carcass, between the 12th and 13th ribs, to measure the thickness of the dorsal fat (mm) and the LD muscle area with a planimeter (Planix 7 TAMAYA). Chop samples were then obtained between the 10th and 13th ribs; these samples were vacuum-packed and frozen at −20 °C for further laboratory analysis.

## 2.3. Meat Quality Assessment
Nutritional Composition, Tenderness, Water Retention Capacity and Maturation (pH and Color)

LD muscle samples were used to assess the quality of the meat, which were thawed slowly in the refrigerator and then at room temperature. The nutritional composition of the meat was evaluated according to AOAC [39] procedures: dry mass (DM) (2001.12) determined by post-drying weight loss in an oven at 102 °C, crude protein (CP) by the Kjeldahl method (968.06), ether extract (EE) by the Soxhlet method (920.39) and ash by incineration in a muffle oven at 550 °C (935.12). Chop samples were thawed to assess meat tenderness, and sections of LD muscle were used to measure cutting force with a Warner-Bratzler blade (SALTE R, G-R® Elec. Mfg. Co., Collins Lane, MA, USA). Once

the meat was thawed, it was cooked on an electric grill to 70 °C internal temperature and left at room temperature; cylinders 2.5 cm long × 1.27 cm diameter were cut with a drill (Mod. 14658, Truper®, Mexico). The meat cylinders were cut with a Warner-Bratzler knife, recording the cutting force data (kgf/cm²) [45]. To assess the water-holding capacity (WHC) in chilled fresh meat, a chop sample (50 g) was taken for immediate determination for 24 h by the gravimetric method [46].

The meat maturation was evaluated in chop samples collected between the 12th and 13th ribs. The pH was measured at 24 h, 4 and 8 d after slaughter with a potentiometer equipped with a penetration electrode (HANNA model HI 99163). Color parameters were measured with a calibrated colorimeter (Minolta Croma Meter CR-400, Osaka, Japan) according to Hunt et al. [47] in a measuring area of 10 mm diameter at room temperature (approximately 15 to 20 °C) with D65 illuminant and a viewing angle of 0°, to detect differences in the redness of meat samples between applied treatments. Before each measurement, the colorimeter was calibrated in the color space system [48] using a white mosaic. One measurement consisted of three consecutive flashes of illumination to obtain an average value. The parameters of luminosity ($L^*$), reddish ($a^*$ red ± green), yellowish ($b^*$ yellow ± blue), chroma ($C^*$) and hue ($H°$) [49] were recorded.

## 2.4. Analysis of Fatty Acids

The fatty acid content was determined by gas chromatography according to the methodology described by Rodríguez-Maya et al. [24]; 1 μL of each sample was injected into the gas chromatograph (Perkin Elmer, model Clarus 500), the fatty acids were separated in a capillary column of 100 m × 0.25 mm inner diameter × 0.2 μm film thickness (SUPELCO TM-2560) and the separation was obtained by temperature ramp (140 °C for 5 min, with increases of 4 °C per min to 240 °C), using nitrogen as a carrier gas. Retention times were compared with known standards (SUPELCO37, SIGMA USA analytical FAME MIX). The saturation (S/P), iatrogenic (IAI) and teratogenic (TRI) indices were calculated as described by Enser et al. [50] and Ulbricht and Southgate [51]:

$$P/S = (C18:2w6 + C18:2w3)/(C14:0 + C16:0 + C18:0);$$

$$IAI = (C12:0 + 4 \times C14:0 + C16:0)(/\sum SFA + \sum PUFA);$$

$$TRI = (C14:0 + C16:0 + C18:0)/[0.5 \times (\sum SFA + 0.5) \times (\sum(n-6) + 3 \times \sum(n-3))/(\sum(n-3)/\sum(n-6))]$$

where SFA is the total saturated FA and PUFA is the total polyunsaturated FA.

## 2.5. Total RNA Extraction and Reverse Transcription

Immediately after slaughtering the sheep, a sample in each left carcass of the LD muscle at the level of the 13th rib was collected, frozen in liquid nitrogen and stored at −80 °C until analysis. Total RNA extraction from tissue samples was performed using the Trizol technique using a commercial kit (Molecular Research Center, Cincinnati, OH, USA) according to the manufacturer's protocol and specifications. RNA concentration was quantified by measuring absorbance at 260 nm, and RNA integrity was measured by total RNA electrophoresis using a 1% agarose formaldehyde gel, stained with ethidium bromide, for visualization of ribosomal RNA in the 28 s and 18 s subunit bands. From the total RNA, the retro-transcription reaction (RT) was performed to produce the complementary DNA of the first strand (cDNA) using the iScript cDNA synthesis kit (Bio-Rad, Hercules, CA, USA) according to the manufacturer's specifications.

## 2.6. Primers and Real-Time PCR Quantification

Six genes classified and involved in intermediate lipid metabolism were selected and frequently used as a reference in real-time PCR gene expression experiments (qRT-PCR). The oligonucleotides selected for real-time analysis were those reported by Jeong et al. [52]. The sequence of primers and estimated size of PCR products are shown in Table 2. The PCR amplification was performed with ABI PRISM 7500 (Applied Biosystems, USA) equipment,

using the technique of 96 optical well plates with SYBR Green PCR Master Mix. Recognition temperatures were optimized for individual genes and primers. Standard curves were made to calculate amplification efficiency during qRT-PCR.

**Table 2.** Oligonucleotides used in quantitative reverse real-time CRP analyses of transcription (qRT-PCR).

| Gene/Symbol | Gene Bank ID | $5'\rightarrow 3'$ | Sequence | Amplicon Size, bp |
|---|---|---|---|---|
| Acetyl-coenzyme A carboxylase alpha, ACC | NM_174224 | Forward | aggagggaagggaatcagaa | 69 |
| | | Reverse | gcttgaacctgtcggaagag | |
| Lipoprotein lipase, LPL | NM_001075120 | Forward | acttgccacctcattcctg | 119 |
| | | Reverse | acccaactctcatacattcctg | |
| Glycerol-3-phosphate Acyltransferase 1, GPAT1 | NM_001012282 | Forward | tgtgctatctgctctccaatg | 116 |
| | | Reverse | ctccgccactataagaatg | |
| Diacylglycerol acyltransferase 1, DGAT1 | NM_174693 | Forward | tcttccactcctgcctgaac | 96 |
| | | Reverse | agtaggtgatggactcggag | |
| Hormone-sensitive lipase, HSL | NM_001080220 | Forward | gatgagagggtaattgccg | 100 |
| | | Reverse | ggatggcaggtgtgaact | |
| Noglyceride lipase, MGL | NM_001163689 | Forward | cactttttcaaggtcttcgctg | 110 |
| | | Reverse | gatgtccacctccgtcttattc | |
| Ribosomal protein S9, RPS9 [1] | NM_001101152 | Forward | cctcgaccaagagctgaag | 64 |
| | | Reverse | cctccagacctcacgttgttc | |

[1] Housekeeping gene? Jeong et al. [52].

### 2.7. Gene Expression Analysis

To analyze the expression pattern of the six selected genes, the reactions of the qRT-PCR started at 95 °C for 5 min for pre-denaturation; the conditions were set at 95 °C for 10 s, 60 °C for 35 s and 72 °C for 30 s. The PCR cycle performed 40 cycles. The relative quantification of the abundance expression of the selected genes was evaluated with the $^{\Delta\Delta}C_T$ method. All data were normalized using a housekeeping gene (ribosomal protein S9). The double change in the relative expression of the target genes was determined by calculating the $2^{-\Delta CT}$.

### 2.8. Statistical Analysis

The data of this study were statistically analyzed with the PROC MIXED procedure [53] under a completely randomized design with $2 \times 2$ factorial arrangement of treatments (-ZH: 0 mg kg$^{-1}$ LW or +ZH: 0.2 mg kg$^{-1}$ LW; -ZM: 0 mg Zn kg$^{-1}$ DM or +ZM: 80 mg Zn kg$^{-1}$ DM), with nine lambs per treatment, using a mixed model that included the lamb effect (random subject), level of ZH and ZM supplementation (fixed) and residual (lamb within supplementation with HZ and ZM) with ANTE covariance structure (1). DWG and DMI variables and pH and coloration parameters at the meat maturation study were analyzed using the same model, including the time effect (repeated) and time interaction with ZH and ZM supplementation level in the model. Categorical data (conformation, fat degree and perirenal fat) were analyzed using the SAS NPAR1WAY procedure [53]. The means of treatments with significant difference ($p \le 0.05$) were compared with Tukey's test.

## 3. Results

### 3.1. Productive Performance

Sheep growth (Table 3) in terms of FLW, DMI, TWG and DWG was not affected ($p > 0.05$) by the treatments applied; however, a tendency ($p = 0.076$) to change in sheep treated with ZH was observed in the variable FC. Productive performance was not affected by the ZM and there was no effect of ZM interaction ($p > 0.05$) with ZH.

**Table 3.** Productive performance of sheep supplemented with zilpaterol hydrochloride and zinc methionine.

| Variables | Treatments | | | | [1] SEM | Effect | | |
|---|---|---|---|---|---|---|---|---|
| | −ZH | | +ZH | | | ZH | ZM | ZH × ZM |
| | −ZM | +ZM | −ZM | +ZM | | | | |
| ILW (kg) [2] | 25.375 | 25.150 | 25.400 | 25.150 | 2.053 | —— | —— | —— |
| FLW (kg) [3] | 44.300 | 44.100 | 44.975 | 45.500 | 1.395 | Ns | ns | ns |
| TGW (kg) [4] | 18.925 | 18.950 | 19.575 | 20.375 | 1.548 | Ns | ns | ns |
| DMI (kg d$^{-1}$) [5] | 1.436 | 1.308 | 1.349 | 1.331 | 0.102 | Ns | ns | ns |
| DWG (kg d$^{-1}$) [6] | 0.253 | 0.255 | 0.264 | 0.275 | 0.021 | Ns | ns | ns |
| FC (kg) [7] | 5.676 | 5.131 | 5.109 | 4.839 | 0.287 | 0.076 [Ł] | ns | ns |

−ZH, zilpaterol hydrochloride 0 mg kg$^{-1}$ LW; +ZH, zilpaterol hydrochloride 0.2 mg kg$^{-1}$ LW; −ZM, zinc methionine 0 mg kg$^{-1}$ DM; +ZM, zinc methionine 80 mg kg$^{-1}$ DM. [1] SEM, standard error of the mean. [Ł] Statistical trend ($p > 0.05$ to $p \leq 0.10$). ns, not significant. [2] ILW, initial live weight; [3] FLW, final live weight; [4] TWG, total weight gain; [5] DMI, dry matter intake; [6] DWG, daily weight gain; [7] FC, feed conversion.

### 3.2. Carcass Traits

Carcass traits improved with ZH (Table 4); there was an increase ($p \leq 0.074$) in the variables HCW ($p \leq 0.074$), CCW and CMUE ($p \leq 0.065$), HCY, CCY, CI, greasing, chop area and crude protein ($p \leq 0.05$). The addition of ZM did not affect any of the traits described above, but the content of ether extract in the intramuscular fat of sheep meat increased ($p \leq 0.001$) due to the effect of ZM and also ($p \leq 0.05$) the interaction of the treatment ZH with ZM (Table 4).

**Table 4.** Carcass traits and quality attributes of meat from sheep supplemented with zilpaterol hydrochloride and zinc methionine.

| Variables | Treatments | | | | [1] SEM | Effect | | |
|---|---|---|---|---|---|---|---|---|
| | −ZH | | +ZH | | | ZH | ZM | ZH × ZM |
| | −ZM | +ZM | −ZM | +ZM | | | | |
| LWS (kg) [2] | 42.000 | 40.925 | 42.425 | 42.687 | 0.975 | ns | ns | ns |
| HCW (kg) [3] | 19.687 | 20.587 | 21.675 | 22.212 | 0.860 | 0.074 [Ł] | ns | ns |
| CCW (kg) [4] | 18.675 | 18.900 | 20.075 | 20.800 | 1.418 | 0.065 [Ł] | ns | ns |
| HCY (%) [5] | 46.762 | 50.287 | 51.075 | 51.887 | 1.303 | 0.046 * | ns | ns |
| CCY (%) [6] | 44.275 | 46.175 | 47.287 | 48.700 | 0.490 | 0.042 * | ns | ns |
| LC (cm) [7] | 62.440 | 64.370 | 62.630 | 62.750 | 1.965 | ns | ns | ns |
| CI [8] | 0.298 | 0.293 | 0.317 | 0.331 | 0.011 | 0.024 * | ns | ns |
| CMUE (1–5) [9] | 3.100 | 3.100 | 3.500 | 3.600 | 0.230 | 0.065 [Ł] | ns | ns |
| Greasing (1–5) [10] | 3.000 | 3.100 | 2.700 | 2.500 | 0.110 | 0.014 * | ns | ns |
| Chop area (cm$^2$) | 12.30 | 15.400 | 17.400 | 16.800 | 1.310 | 0.020 * | ns | ns |
| Back fat (mm) | 2.000 | 3.100 | 2.200 | 2.200 | 0.970 | ns | ns | ns |
| Kidney fat (1–4) [11] | 1.800 | 2.000 | 1.800 | 1.800 | 0.960 | ns | ns | ns |
| Dry matter (g/100 g) | 28.800 | 30.500 | 29.400 | 29.600 | 0.690 | ns | ns | ns |
| Ashes (g/100 g) | 1.460 | 1.550 | 1.660 | 1.520 | 0.210 | ns | ns | ns |
| Ether extract (g/100 g) | 4.400 | 6.130 | 4.090 | 6.920 | 0.690 | ns | 0.001 | 0.043 * |
| Crude protein (g/100 g) | 22.200 | 22.000 | 23.700 | 23.700 | 0.820 | 0.049 * | ns | ns |
| Water loss 24 h (g/kg$^{-1}$) | 8.150 | 9.240 | 7.930 | 7.950 | 0.47 | ns | ns | ns |
| Cutting force (kg/cm$^2$) | 7.880 | 8.390 | 10.700 | 9.850 | 0.64 | ns | ns | ns |

−ZH, zilpaterol hydrochloride 0 mg kg$^1$ LW; +ZH, zilpaterol hydrochloride 0.2 mg kg$^{-1}$ LW; −ZM, zinc methionine 0 mg kg$^{-1}$ DM; +ZM, zinc methionine 80 mg kg$^{-1}$ DM. [1] SEM, standard error of the mean. [2] LWS, live weight at slaughter; [3] HCW, hot carcass weight; [4] HCY, hot carcass yield; [5] CCW, cold carcass weight; [6] CCY, cold carcass yield; [7] LC, long carcass; [8] compactness index (CCW$_{Kg}$/LC$_{cm}$); [9] muscle conformation and [10] degree of fat European System; [11] internal fat coverage (1, kidney uncovered; 2, large window; 3, small window; 4, kidney cover). *, Significant difference (* $p \leq 0.05$); [Ł] statistical trend ($p > 0.05$ to $p \leq 0.10$). ns, not significant.

### 3.3. pH Values and Meat Color

In the current study, pH values at 24 h post-slaughter were higher ($p \leq 0.001$) in the sheep carcasses with ZH; however, on days 4 and 8 post-slaughter, the pH values were higher ($p < 0.001$) in sheep carcasses with ZM and without ZH. No statistical difference ($p > 0.05$) was observed in the pH values between measurement times. The values of color indices $L^*$, $a^*$, $b^*$, $C^*$ and $H°$ were within the normal range for fattening sheep meat but were lower ($p \leq 0.01$) because of ZH. The ZM did not affect the meat coloration indices described above; there was no effect of interaction ($p > 0.05$) of ZH with ZM (Table 5).

**Table 5.** Effect of zilpaterol hydrochloride and zinc methionine on meat maturation (pH and color values at 24 h, 4 d and 8 d) of sheep in intensive fattening.

| Variables | Treatments | | | | [1] SEM | Effect | | |
|---|---|---|---|---|---|---|---|---|
| | −ZH | | +ZH | | | ZH | ZM | ZH × ZM |
| | −ZM | +ZM | −ZM | +ZM | | | | |
| pH at 24 h | 5.62 | 5.85 | 6.13 | 6.06 | 0.637 | 0.001 ** | ns | ns |
| pH at 4 d | 5.66 | 6.60 | 6.13 | 6.00 | 0.107 | 0.001 ** | ns | ns |
| pH at 8 d | 5.71 | 6.60 | 6.18 | 6.11 | 0.114 | 0.001 ** | ns | ns |
| Color | | | | | | | | |
| $L_{24h}$ | 41.1 | 39.2 | 35.4 | 35.8 | 1.878 | 0.019 * | ns | ns |
| $a_{24h}$ | 16.1 | 14.2 | 12.2 | 12.0 | 1.387 | 0.004 ** | ns | ns |
| $b_{24h}$ | 6.15 | 5.38 | 3.6 | 3.55 | 0.577 | 0.001 ** | ns | ns |
| $C_{24h}$ | 17.2 | 16.4 | 12.7 | 12.6 | 1.071 | 0.001 ** | ns | ns |
| $H°_{24h}$ | 21.5 | 18.8 | 14.2 | 16.3 | 2.452 | 0.009 ** | ns | ns |
| $L_{4d}$ | 39.7 | 39.6 | 33.3 | 33.5 | 1.444 | 0.001 ** | ns | ns |
| $a_{4d}$ | 15.2 | 13.9 | 11.3 | 11.8 | 0.982 | 0.002 ** | ns | ns |
| $b_{4d}$ | 9.09 | 8.99 | 5.46 | 5.30 | 0.976 | 0.001 ** | ns | ns |
| $C_{4d}$ | 17.8 | 16.6 | 12.6 | 13.0 | 1.190 | 0.010 ** | ns | ns |
| $H°_{4d}$ | 30.6 | 30.5 | 21.3 | 21.1 | 3.573 | 0.001 ** | ns | ns |
| $L_{8d}$ | 38.1 | 38.9 | 34.0 | 33.0 | 1.999 | 0.010 ** | ns | ns |
| $a_{8d}$ | 11.9 | 11.7 | 10.6 | 10.1 | 1.703 | ns | ns | 0.091 [Ł] |
| $b_{8d}$ | 9.78 | 9.54 | 5.39 | 5.08 | 0.777 | 0.001 ** | ns | ns |
| $C_{8d}$ | 15.7 | 15.2 | 12.1 | 12.0 | 0.871 | 0.004 ** | ns | 0.082 [Ł] |
| $H°_{8d}$ | 39.4 | 39 | 27.6 | 27.4 | 2.389 | 0.001 ** | ns | ns |

−ZH, zilpaterol hydrochloride 0 mg kg$^{-1}$ LW; +ZH, zilpaterol hydrochloride 0.2 mg kg$^{-1}$ PV; −ZM, zinc methionine 0 mg kg$^{-1}$ DM; +ZM, zinc methionine 80 mg kg$^{-1}$ DM. [1] SEM, standard error of the mean. *, ** Significant difference (* $p \leq 0.05$; ** $p \leq 0.01$); [Ł] statistical trend ($p > 0.05$ to $p \leq 0.10$). ns, not significant.

### 3.4. Fatty Acids

As shown in Table 6, the content in IMF of stearic (C18:0) and arachidic (C20:0) FA was reduced ($p \leq 0.05$) by the effect of ZH, while that of the FAs palmitoleic (C16:1), eicosatetraenoic (C20:4n6) and conjugated linoleic were increased ($p \leq 0.05$) by ZH. Table 7 shows that ZH tended to increase the content of PUFA ($p \leq 0.062$) and omega-6 FA ($p \leq 0.062$), as well as other PUFA ($p \leq 0.058$). The supply of ZM without ZH increased ($p \leq 0.05$) IMF in meat, the content of palmitoleic FA (C16:1) and conjugated linoleic fatty acid; the ZH interaction with ZM increased ($p \leq 0.05$) the content of linoleic (C18:2c9c12), linolenic (C18:3c9c12c15) and eicosatetraenoic (C20:4n6) FA content. The interaction of ZH with ZM was also observed (Table 7) by reducing ($p \leq 0.05$) the content of SFA ($p \leq 0.05$) and increasing that of UFA and PUFA ($p \leq 0.05$). The iatrogenic and teratogenic indices were not affected ($p > 0.05$) by the supply of ZH and ZM to fattening sheep.

**Table 6.** Effect of zilpaterol hydrochloride and zinc methionine on the content of fatty acids in intramuscular fat of Longissimus dorsi muscle of sheep in intensive fattening.

| Fatty Acids | Treatments | | | | [1] SEM | Effect | | |
|---|---|---|---|---|---|---|---|---|
| | −ZH | | +ZH | | | ZH | ZM | ZH × ZM |
| | −ZM | +ZM | −ZM | +ZM | | | | |
| C10:0, Caproic | 0.128 | 0.130 | 0.155 | 0.113 | 0.018 | ns | ns | ns |
| C12:0, Lauric | 0.130 | 0.193 | 0.203 | 0.161 | 0.034 | ns | ns | 0.050 * |
| C14:0, Myristic | 2.393 | 2.981 | 2.948 | 2.93 | 0.278 | ns | ns | ns |
| C14:1 n-5, myristoleic | 0.301 | 0.288 | 0.363 | 0.296 | 0.034 | ns | ns | ns |
| C15:0, Pentadecanoic | 0.705 | 0.553 | 0.728 | 0.736 | 0.114 | ns | ns | ns |
| C16:0, Palmitic | 25.71 | 26.26 | 25.78 | 24.80 | 0.723 | ns | ns | ns |
| C16:1, Palmitoleic | 1.235 | 1.530 | 1.601 | 1.605 | 0.095 | 0.009 ** | 0.050 * | 0.073 [Ł] |
| C17:0, Heptadecanoic | 0.886 | 0.753 | 0.803 | 0.746 | 0.079 | ns | ns | ns |
| C17:1, Heptadecenoic | 0.566 | 0.571 | 0.586 | 0.606 | 0.067 | ns | ns | ns |
| C18:0, Stearic | 18.26 | 17.01 | 17.3 | 16.72 | 1.089 | 0.048 * | ns | ns |
| C18:1$n9c$, Oleic | 41.56 | 41.84 | 40.72 | 41.67 | 1.291 | ns | ns | ns |
| C18:2 c9c12, Linoleic | 5.785 | 5.905 | 6.461 | 6.963 | 1.911 | ns | ns | 0.050 * |
| C18:3 c 9c12c15, Linolenic | 0.033 | 0.041 | 0.05 | 0.058 | 0.01 | ns | ns | 0.050 * |
| C20:0, Arachidic | 0.038 | 0.036 | 0.031 | 0.016 | 0.009 | 0.050 * | ns | ns |
| C20:4n6, Eicosatetraenoic | 0.986 | 1.175 | 1.613 | 1.971 | 0.312 | 0.011 * | ns | 0.049 * |
| C22:0, Behenic | 0.086 | 0.151 | 0.15 | 0.211 | 0.039 | 0.060 [Ł] | 0.061 [Ł] | ns |
| Other trans acids | 1.181 | 0.565 | 0.491 | 0.39 | 0.294 | 0.081 [Ł] | 0.101 [Ł] | ns |
| C18:1 n-7, Vaccenic acid | 1.063 | 0.42 | 0.33 | 0.24 | 0.286 | 0.060 [Ł] | 0.101 [Ł] | ns |
| Conjugated linoleic acid | 0.115 | 0.143 | 0.161 | 0.17 | 0.029 | 0.045 * | 0.031 * | ns |

−ZH, zilpaterol hydrochloride 0 mg kg$^{-1}$ LW; +ZH, zilpaterol hydrochloride 0.2 mg kg$^{-1}$ LW; −ZM, zinc methionine 0 mg kg$^{-1}$ DM; +ZM, zinc methionine 80 mg kg$^{-1}$ DM. [1] SEM, standard error of the mean. *, ** Significant difference (* $p \leq 0.05$; ** $p \leq 0.01$); [Ł] statistical trend ($p > 0.05$ to $p \leq 0.10$). ns, not significant.

**Table 7.** Effect of zilpaterol hydrochloride and zinc methionine on the fatty acid content and iatrogenic and teratogenic indices of intramuscular fat of Longissimus dorsi muscle of sheep in intensive fattening.

| Fatty Acids | Treatments | | | | [1] SEM | Effect | | |
|---|---|---|---|---|---|---|---|---|
| | −ZH | | +ZH | | | ZH | ZM | ZH × ZM |
| | −ZM | +ZM | −ZM | +ZM | | | | |
| SFA [2] | 48.350 | 48.082 | 48.112 | 46.447 | 1.718 | ns | ns | 0.048 * |
| UFA [3] | 52.635 | 53.003 | 52.980 | 54.713 | 1.582 | ns | ns | 0.050 * |
| MFA [4] | 44.730 | 44.650 | 43.603 | 44.420 | 1.324 | ns | ns | ns |
| PUFA [5] | 7.067 | 7.422 | 8.462 | 9.335 | 1.185 | 0.06 | ns | 0.049 * |
| Ω-3FA | 0.033 | 0.042 | 0.050 | 0.048 | 0.009 | ns | ns | ns |
| Ω-6FA | 6.772 | 7.082 | 8.075 | 8.937 | 1.158 | 0.061 [Ł] | ns | ns |
| IAI [6] | 0.542 | 0.565 | 0.573 | 0.540 | 0.049 | ns | ns | ns |
| TRI [7] | 1.000 | 0.990 | 1.005 | 0.938 | 0.064 | ns | ns | ns |

−ZH, zilpaterol hydrochloride 0 mg kg$^{-1}$ LW; +ZH, zilpaterol hydrochloride 0.2 mg kg$^{-1}$ PV; −ZM, zinc methionine 0 mg kg$^{-1}$ DM; +ZM, zinc methionine 80 mg kg$^{-1}$ DM. [1] SEM, standard error of the mean. *, Significant difference (* $p \leq 0.05$); [Ł] statistical trend ($p > 0.05$ to $p \leq 0.10$). ns, not significant. [2] SFA = saturated FA, [3] UFA = unsaturated FA, [4] MFA = monounsaturated FA, [5] PUFA = polyunsaturated FA, [6] IAI = iatrogenic index, [7] TRI = teratogenic index.

### 3.5. Gene Expression

In the present study, the supply of ZM and ZH affected ($p \leq 0.05$) the abundance of relative mRNA expression of lipid metabolism genes in LD (Figure 1). ZH affected ($p \leq 0.05$) the relative expression of mRNA of the LPL, HSL, GPAT1 and DGAT1 enzyme gene, expressed, respectively, 0.51, 0.14, 0.42 and 0.89 times more than the group without ZH. ZM affected ($p \leq 0.05$) the relative expression of mRNA of the ACC, HSL, MGL and DGAT1 enzyme genes, expressing, respectively, 1.30, 0.11, 0.69 and 0.61 times more than the group without ZM. The interaction ZH with ZM increased ($p \leq 0.05$) the relative expression

of mRNA of the genes of the enzymes HSL and ACC, expressing, respectively, 0.90 and 0.32 times more than the control group.

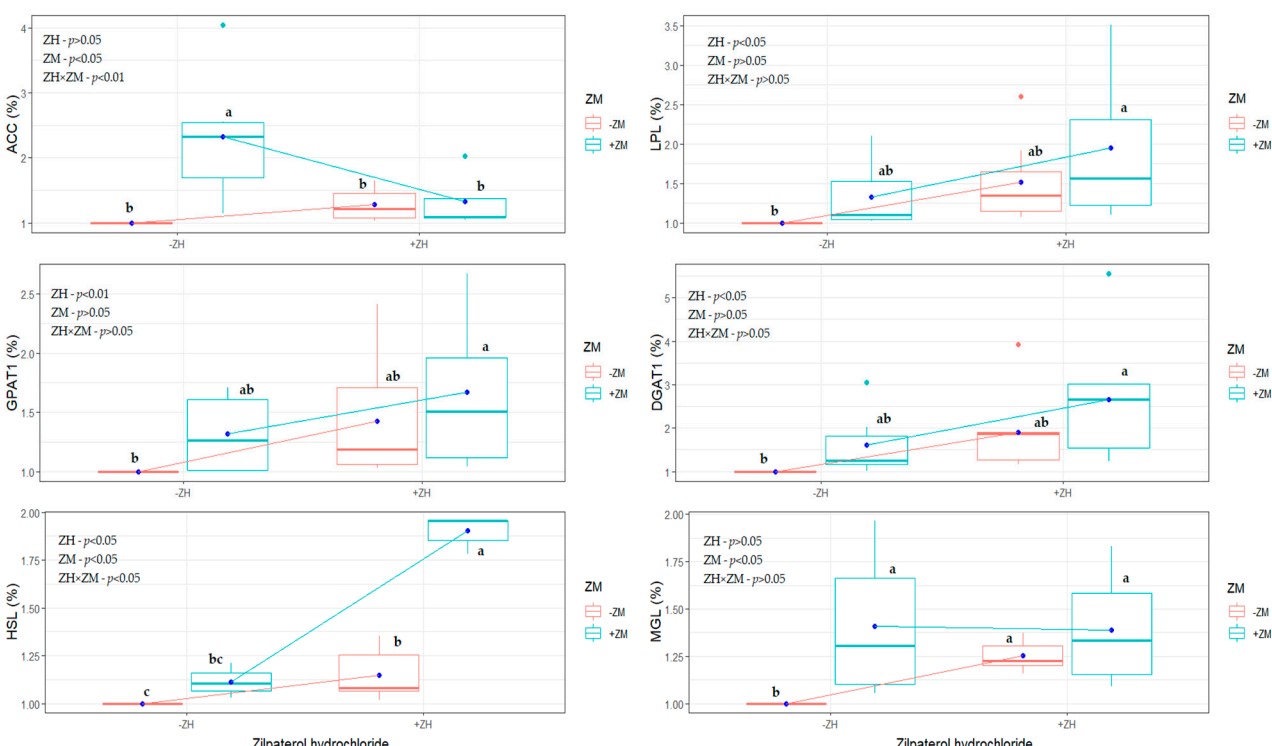

**Figure 1.** Relative expression of mRNA enzymes acetyl CoA carboxylase (ACC), lipoprotein lipase (LPL), glycerol 3 phosphate acyltransferase (GPAT1) and diglyceride acyl transferase (DGAT1), sensitive hormone lipase (HSL) and monoglyceride lipase (MGL), in muscle fat Longissimus dorsi of intensive fattening sheep supplemented with zilpaterol hydrochloride (ZH: 0 or 0.2 mg kg$^{-1}$ LW) and zinc methionine (ZM: 0 or 80 mg kg$^{-1}$ DM). a,b,c, Different letters in different boxes indicate significant difference ($p < 0.05$; $p < 0.01$).

## 4. Discussion

In general, the use of βAAs may increase growth and feed efficiency [8] by stimulating protein synthesis in skeletal muscle and preventing its degradation, while in adipose tissue, βAAs increase lipolysis and reduce lipogenesis [31]. However, there are variations in the response to βAA supply. Salinas et al. [54] indicated that in Pelibuey sheep, ZH did not improve DGW; similarly, Mondragón et al. [18] concluded that ZH in finishing sheep did not affect the productive performance but did increase LD area.

In the present study, the FC in all treatments was less than 6:1 (kg of ingested DM/kg of live weight gain); this means that the ingested nutrients were deposited efficiently; however, in sheep with ZH, the best FC indicated a more efficient use of nutrients for DWG, with more significant feed savings (7.94 %) compared to those that did not receive ZH. In a similar study, Dávila-Ramírez et al. [55] found that fattening female sheep with ZH did not improve FC (7.04 vs. 7.40 kg) ($p > 0.08$). According to information analyzed by Mersmann [56], in sheep, the expected average improvement in FC is 15 %, a value higher than that of the present study (7.94 %); in this regard, the same author indicated that the greater the selection for growth rate, the response to βAA supply might be less because the animals are approaching their biological maximum growth rate.

In the present study, ZM did not affect the productive response, similar to what Rodríguez-Gaxiola et al. [12] observed in fattening cattle supplemented with ZM; in contrast, Rodríguez-Maya et al. [24] observed that the supply of ZM improved DWG and FC in fattening sheep. This may be because the sheep covered their requirements (24 to

51 mg Zn kg$^{-1}$ DM) for the physiological stages of growth and finishing as established by the NRC [35].

The results of the present study on the evaluation of the carcass, in which the ZH increased the weight, yield, compactness index, area of the chop and muscle conformation in 8.80, 6.12, 9.83, 26.4 and 14.52%, respectively, and reduced the degree of fattening of the cold carcass by 14.75%, coinciding with what was observed by López et al. [57] and Mondragón et al. [18], which may be because of ZH on muscle protein synthesis [58]; in addition, changes have been observed in the proportion of transcription RNA in muscle proteins [59] and transcripts with differential expression in skeletal muscle due to ZH (Kubik et al. [60]). Carcass fat decreased ($p \leq 0.05$) due to the effect of ZH, similar to the study of Koohmaraie et al. [61], producing leaner sheep carcasses; this may be as a result of the lipolytic effect of ZH on the adipocyte, which limits and reduces lipogenesis [62]. According to the information collected and analyzed by Mersman [56], on average, βAAs such as ZH can increase muscle mass in fattening cattle and sheep by 10 and 25%, and reduce the fat content in the carcass by 30 and 25% in the same order. Recent results indicate that ZH reduced carcass fat by 10.24% in cattle [14], and in sheep reduced carcass backfat by 7.56 and 5.13% [55,63].

According to Chen et al. [32], zinc supplementation increased body fat deposition in obese mice; Greene et al. [33] reported a higher marbling rate and higher carcass quality of steers fed Zn methionine (360 mg Zn kg$^{-1}$ DM) compared to steers fed ZnO (360 mg kg$^{-1}$ DM) or steers fed with a control diet (82 mg Zn kg$^{-1}$ DM). McBeth et al. [64] found no significant difference in the marbling index of steer meat fed a diet with 90 mg Zn kg$^{-1}$ DM; however, the numerical difference was more than 4% in steers with Zn. Zn has a lipogenic effect [65]; this and its influence on the increase of lipogenesis in adipocytes is mainly due to an increase in insulin activity but not to an insulin-like action [66]; in addition, Zn suppresses the action of nitric oxide [31], which inhibits the synthesis of glucose, glycogen and fats, but stimulates lipolysis in adipocytes, acting by different pathways, through the phosphorylation of adenosine-3'-5'-monophosphate, activated by protein kinases (AMPK), phosphorylating hormone-sensitive lipase (LSH) and perilipins [67]. Regarding the interaction of ZM with ZH, no reports were found in the published literature prior to that indicated by Rodríguez et al. [16], who carried out a study with fattening cattle, concluding that the reduction of intramuscular fat and the thickness of back fat caused by the effect of ZH plus ZM is not very clear. Protein content increased ($p < 0.05$) with the addition of ZH; this was also reported by Partida et al. [68], and attributed to an increase in muscle nitrogen retention induced by ZH [69].

In general, in the present study, the pH values between treatments varied little, only in sheep meat with ZH, at 24 h post slaughter, the pH increased 6.28% concerning to those that did not receive ZH. The pH values obtained (Table 5) coincide with other recent studies carried out in sheep [55,68,69]. The higher values of pH (6.13 and 6.06) in carcasses at 24 h post-mortem, associated with the supply of βAA, may be due to a lower deposition of muscle glycogen affecting the decrease in pH, which usually occurs post-slaughter, because of anaerobic metabolism; in this way, when the pH does not reach a value lower than 6, the meat may have a darker color [70].

The results of the present work regarding the values of the parameter L, which was reduced by the effect of ZH, coincide with those obtained by Choi et al. [14], Brand et al. [63] and Dávila-Ramírez et al. [55]. Likewise, the parameters a, *b* and c of the present study coincide with those reported by Dávila-Ramírez et al. [55] in fattening sheep. The lower coloration values L*, a*, b*, C* and H° associated with the effect of ZH may be due to factors related to the pH value of 24 h post-mortem [71], as well as the different greasing and IMF content in sheep meat with ZH, ZM and ZH plus ZM. Color is an essential factor in selecting meat for the consumer, and in the maturation process, meat color stability is a function primarily of myoglobin-reducing activity and the rate of muscle oxygen consumption. According to Khliji et al. [72], the changes in the degree of redness, determined by the value of the parameter a, are related to consumer acceptance. In the present study, all the

color parameters evaluated at the different measurement times were decreased by ZH. This may be due to the pigmentation of the heme group and a higher proportion of fast glycolytic fibers or to a dilution of myoglobin content in the muscle caused by muscle fiber hypertrophy (Dávila-Ramírez et al. 2016) [55].

In the present work, the content of several of the FAs analyzed in the IMF of the meat was affected by the ZH; these results are different from those observed in the study by Dávila-Ramírez et al. [55] in fattening female sheep, in which ZH reduced ($p \leq 0.05$) the content of eicosapentaenoic FA (C20:5n3), docosahexaenoic (C22:6n-3) and total omega-3 FAs in the LD muscle, but without altering the deposition of body fat in the carcass. In fattening cattle [73], they observed an increase in oleic FA associated with more marbling in the LD muscle. On the other hand, the composition of FA in the LD muscle of fattening cattle was not altered by ZH, nor was the content of saturated and polyunsaturated FA affected [14]. Fritsche et al. [74] reported that ZH supplemented to fattening steers increased the total content of SFA, FA n-3 and FA n-6. Likewise, Ibrahim et al. (2006) [75] observed an increase in the relative proportion of SFA in steers supplemented with the βAA zearalanol. In fattening pigs, in general, studies that have evaluated the effect of ractopamine hydrochloride on the FA profile of meat indicates only minor effects [76,77]; in a study by Carr et al. [78], ractopamine hydrochloride only increased linoleic FA content.

In cattle, pigs and sheep, the main metabolic effects of βAA are to increase muscle mass and decrease adipose tissue in the carcass [56]. βAAs stimulate triacylglycerol degradation in adipocytes and inhibit fatty acid and triacylglycerol synthesis in cells and tissues in vitro of various livestock species; however, it has been reported that, in certain adipocytes of particular livestock species, and with specific βAAs, the results are sometimes negative. Even βAAs that reduce carcass fat and bind to the receptor may have minimal effects on adipocyte lipid metabolism as measured by in vitro assays with βAA itself [79,80]. The increase in plasma of non-esterified fatty acids (NEFAs), after ingestion of βAA, is evidence of lipolysis activated. Several βAAs acutely increase NEFAs in the plasma of pigs [81] and cattle [82]. However, the response can be inhibited by chronically supplementing βAAs in cattle and sheep [82,83].

The information from several studies evaluating the effect of βAAs on the FA content of meat from ruminants and fattening pigs shows different responses in different FAs depending on the animal species; this can be explained by the effect of factors related to the environment, sex, growth rate and maturity of the animal, hormone biosynthesis and ingested diet [84].

Unlike ZH, ZM only influenced the content of palmitoleic (C16:1) and conjugated linoleic FA; by contrast, Rodríguez-Maya et al. [24] observed that the content of myristic acid (C14:0) was lower ($p \leq 0.05$) and that palmitoleic acid (C16:1) and arachidonic acid (C20:4) were higher ($p \leq 0.05$) in the IMF of the LD muscle of lambs supplemented with ZnO and ZM. For human consumption, it is not recommended that the IMF of sheep meat has high levels of myristic and palmitic acids, given their association as a risk factor in cardiovascular disease [85]. Regarding palmitoleic acid, the results of the present study coincide with those obtained by Rodríguez-Maya et al. [24], observing that, in both studies, the ZM increased sheep meat fat content; in this sense, the meat was healthier. Clejan et al. [86] and Cunnane [87] reported that Zn interacts with essential fatty acid metabolism. Zn deficiency also accentuates symptoms of essential fatty acid deficiency [88]. In the present study, the increase in the content of conjugated linoleic acid promoted by the effect of the ZM supplied metabolically is positive, given its importance as a source substrate which, together with linolenic acid, undergoes an elongation in the process of desaturation to carry out the synthesis of arachidonic acid with the action of enzymes $\Delta^6$ and $\Delta^5$ desaturases [89]. Essential polyunsaturated fatty acids, as well as linoleic and linolenic acids, are part of the nutrients in the diet of ruminants; therefore, their content in the diet influences the fatty acid profile of milk or meat fat. It should be noted that in the present study. Iatrogenic and teratogenic indices were not affected by ZH and ZM in fattening sheep; this allows us to deduce that the adipogenicity action of ZM and lipolytic

ZH could promote the synthesis of unsaturated FA in muscle and, therefore, produce healthier meat. However, our study agrees with other researchers that the potential effects of supplementation with the approved BAAs (zilpaterol hydrochloride and ractopamine) on fatty acid content in the meat of livestock species requires further investigation.

In the present study, the evaluation of the effect of ZH and ZM on the relative mRNA expression of the genes of six enzymes related to lipid metabolism in the Longissimus dorsi muscle of intensively fattened sheep showed that ZH influenced mainly three genes; therefore, for the enzymes LPL, GPAT1 and DGAT1, the expression of their respective gene was 50 to 89% higher than in sheep without CZ. This indicates that CZ, through its effect on the expression of these genes, promoted greater absorption, esterification and use of FA through the enzymes mentioned above. ZM also influenced the expression of three genes, in this case for the ACC, MGL and DGAT1 enzymes, so the expression of their respective gene was 130 to 61% higher than in sheep without ZM. This indicates that ZM, through its effect on the expression of these genes, promoted greater lipogenesis and FA esterification through these enzymes. Likewise, only the genes of the ACC and HSL enzymes were expressed 30 and 90% more due to the interaction of ZH with ZM. This could indicate the initial participation of ZM in lipogenesis processes and later of CZ in lipolysis to generate energy for protein synthesis in muscle.

The metabolism of dietary lipids in the rumen determines, to a great extent, the deposition and composition of fat of ruminant muscle; therefore, the synthesis of fat is the main factor affecting the FA composition in the muscle of livestock, so anabolic and catabolic processes regulate the balance between the deposit and removal of fat in the muscle; likewise, the expression of lipid metabolism genes in muscles, including those involved in lipogenesis, FA uptake, FA esterification, lipolysis and FA oxidation, may influence IMF deposition, marbling grade and meat qualities such as juiciness, tenderness and taste [90,91].

In sheep, the expression of mRNA genes of lipid metabolism enzymes varies according to the location of adipose tissue in the carcass. Gallardo et al. [92] observed that, in grazing sheep which did not receive Zn, the relative expression of mRNA of the genes of the lipogenic enzymes ACC and FASN in tail fat, was 5.9 and 3.2 times greater, respectively, in the native Chilota breed vs. the Suffolk Down breed. ACC and FASN enzymes play an essential role in synthesizing triglycerides, a primary energy transport and storage source. Under conditions of high energy consumption, as occurs in the fattening of cattle and sheep with intensive feeding, the ACC converts acetyl CoA into malonyl CoA, this is used by the enzyme FASN to form palmitic FA, capable of being desaturated to palmitoleic FA by stearyl-CoA desaturase (SCD1) or elongated to stearic acid, and this can be desaturated to oleic acid by SCD1 itself [93]. In the present study, the greater expression of the mRNA of the ACC enzyme gene in sheep with ZM coincides with the increase in the IMF of the LD muscle, as well as the content of palmitoleic acid in IMF by the effect of both ZH and ZM, as well as the increase in stearic acid due to the effect of ZH. According to Mwangi et al. [94], in cattle, the gene for the enzymes stearoyl CoA desaturase and FA synthase are correlated with subcutaneous fat deposition and degree of marbling [95]. The FA synthase gene encodes a multifunctional polypeptide of enzymes associated with the biosynthesis of FA in the cytoplasm of animal cells, as well as with subcutaneous fat, chop area and IMF of cattle [2,96]. In fattening cattle, Jeong et al. [52] observed a higher abundance in mRNA expression of nine genes for lipogenesis, lipid uptake and fatty acid esterification, showing positive correlations with IMF content in LD muscle. Their results indicated that the abundance of mRNA for the acetyl-CoA carboxylase (ACC) and fatty acid synthase (FASN) genes was correlated with the IMF content.

The availability of fatty acids influences IMF content. Regulation of the flow of FAs into the adipocytes of the LD muscle is essential to provide the substrates required for IMF deposition. Studies on mRNA expression of fatty acid uptake genes indicate that the LPL gene is correlated with IMF content [52]. LPL is an enzyme limiting the hydrolysis rate of triacylglycerol (TG), circulating lipoproteins rich in TG, chylomicrons and very-low-density

lipoproteins. The products of the LPL-catalyzed reaction, fatty acids and monoacylglycerol, are taken up, in part, by adipose tissues and skeletal muscle and stored as neutral lipids [97]. Previous studies suggest that the relative abundance of LPL within adipose and muscle tissues determines the partition of plasma TG between these tissues [98,99]. In the present study, ZH increased the abundance in LPL gene expression; therefore, as suggested by Jeong et al. [52], this gene may be a genetic marker predictive of IMF deposition.

Concerning enzymes that esterify fatty acids for TG synthesis, the process is considered to favor a higher IMF content. In the present investigation, the abundance of expression of the mRNA of the FA esterification gene, including GPAT1 and DGAT1, showed an effect of ZH in both; this coincides with what was found by Jeong et al. [52], which showed positive correlations with the IMF content. TG synthesis begins with the acylation of glycerol-3-phosphate to form 1-acylglycerol-3-phosphate or lysophosphatidic acid (LPA), a rate-limiting step of synthesis catalyzed by one of several GPAT1 isoforms to form 1-acylglycerol-3-phosphate. Therefore, GPAT1 is a key to IMF deposition. The accumulation of intramyocellular TG due to the increased availability of systemic FAs causes increased GPAT1 activity [100]; therefore, the GPAT1 gene could be used as a genetic marker to predict IMT deposition [52]. For DGAT1 gene expression, lysophosphatidic acid is further accommodated at the sn-2 position by 1-acylglycerol-3-phosphate acyltransferase (AGPATA) to form phosphatidic acid [101]. The phosphate group is then removed by phosphatide phosphohydrolase to produce diacylglycerol (DG). The DGAT1 enzyme catalyzes the final step of TG formation from diacylglycerol and acyl-CoA [102]; increased DGAT1 activity increases triglyceride synthesis in mouse muscle [103].

Lipid removal by lipolysis may also be associated with IMF deposition in the LD muscle. In the present study, mRNA expression of the MGL gene was not affected by ZH interaction with ZM; likewise, the abundance of MGL mRNA and adipose triglyceride lipase (ATGL) showed a negative correlation with the IMF content in LD. The ATGL enzyme hydrolyzes the first ester bond of stored TGs, thereby releasing unesterified free FAs [104]; the above information suggests that decreased MGL and ATGL activities increase meat marbling [52].

The HSL enzyme hydrolyzes diacylglycerol, a product of the action of triglyceride lipase (ATGL), to monoglycerides. In the present study, ZH interaction with ZM increased the expression of the mRNA of the HSL gene; Kazala et al. and Jeong et al. [52,105] indicated that the abundance of mRNA of the HSL gene was associated with bovine IMF content. Positive and negative correlations have been reported between IMF and the activity of HSL and ATGL enzymes, indicating that it may be due to some encapsulation of lipid droplets by perilipin, which would reduce the availability of triacylglycerol for HSL.

## 5. Conclusions

It is concluded that the ZH in fattening sheep improved their feed conversion, increased the yield in carcasses and decreased their degree of greasing. ZM increased IMF in *Longissimus dorsi*. ZH and ZM influenced the composition of fatty acids, reduced SFA and increased UFA and PUFA. Both ZH and ZM influenced the abundance of relative mRNA expression of genes involved in fatty acid metabolism (lipogenesis, uptake, esterification and lipolysis).

The observed effect of zilpaterol clohydrate alone or in interaction with zinc methionine on growth, meat quality, and fatty acid profile and gene expression of lipid metabolism may represent essential advantages in the quality of sensory attributes of meat in intensive fattening sheep.

**Author Contributions:** Conceptualization, I.A.D.-V., J.M.P.-R., J.L.B.-G., D.H.-R., E.M.-A., J.E.S.-T. and M.G.-B.; methodology, I.A.D.-V., J.M.P.-R., D.H.-R., D.T.-G., M.A.R.-G., E.M.-A., J.E.S.-T., F.G.-N. and M.G.-B.; software, I.A.D.-V., D.H.-R., D.T.-G., E.M.-A., J.L.B.-G. and M.G.-B.; validation, I.A.D.-V., D.H.-R., D.T.-G., E.M.-A., J.E.S.-T., M.G.-B., and F.G.-N.; formal analysis, I.A.D.-V., J.M.P.-R., D.H.-R., D.T.-G., E.M.-A., J.E.S.-T., and M.G.-B.; investigation, I.A.D.-V., J.M.P.-R., D.H.-R., J.L.B.-G., D.T.-G., E.M.-A., J.E.S.-T., and M.G.-B.; resources, I.A.D.-V., D.H.-R., D.T.-G., E.M.-A., J.E.S.-T., and M.G.-B.;

data curation, I.A.D.-V., D.H.-R., D.T.-G., E.M.-A. and M.G.-B.; writing—original draft preparation, I.A.D.-V., J.M.P.-R., D.H.-R., D.T.-G., M.G.-B., E.M.-A. and G.V.-G.; writing—review and editing, I.A.D.-V., J.M.P.-R., J.L.B.-G., D.T.-G., F.G.-N. and G.V.-G.; project administration and funding acquisition, I.A.D.-V., J.M.P.-R., D.H.-R., E.M.-A., J.E.S.-T., M.A.R.-G. and F.G.-N. All authors have read and agreed to the published version of the manuscript.

**Funding:** This research was funded by the Autonomous University of the State of Mexico through the research project with registration number 4293/2017/CI/UAEM and by the National Council of Science and Technology of the Government of Mexico through a scholarship to study Master of Science studies.

**Institutional Review Board Statement:** This research was approved by the Bioethics and Animal Welfare Committee of the Faculty of Veterinary Medicine and Zootechnics of the Autonomous University of the State of Mexico.

**Data Availability Statement:** The information published in this study is available on request from the corresponding author.

**Acknowledgments:** To the Faculty of Veterinary Medicine and Zootechnics of the Autonomous University of the State of Mexico, and to the National Council of Science and Technology (CONACyT) of the Government of Mexico, for the facilities and financing granted to carry out the research and grant a scholarship to the master's student.

**Conflicts of Interest:** The authors declare that they do not have any personal or institutional conflicts of interest.

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
