# Peer review of "Effect of Zilpaterol Hydrochloride and Zinc Methionine on Growth, Carcass Traits, Meat Quality, Fatty Acid Profile and Gene Expression in Longissimus dorsi Muscle of Sheep in Intensive Fattening"

_agriculture, doi:10.3390/agriculture13030684_

Round 1

Reviewer 1 Report

The work makes a contribution to the use of the additive under study and methionine in an isolated or associated way on the manipulation of the diet of sheep for weight gain with adequate carcass finishing. There are no rectifications, despite each group of researchers having their own writing style and reasoning for the observed results, regarding the extent and way of working with the data.

Author Response

No hay rectificaciones, a pesar de que cada grupo de investigadores tiene su propio estilo de redacción y razonamiento sobre los resultados observados, en cuanto a la extensión y forma de trabajar con los datos.

Author Response

General comment: In the manuscript, the author studied the effects of Zilatro (ZH) and Zimethionine (ZM) on the production of fatty acid metabolism-related genes and mRNA expression characteristics in sheep.

The description of the result part is incomplete and confusing and must be corrected.

R. The wise opinions and suggestions of the two commentators were discussed in the manuscript, which greatly improved the results chapter.

R. The discussion was expanded, especially with regard to the variables of the impact of supplementary additives.

Specific comments:

L259: LWS and LC are also not affected by treatment.

R. The variables mentioned by the reviewer (LWS and LC) have been included in the text.

L262: PCF is not shown in Table 3.

R. The variables indicated by the reviewer have been corrected in the manuscript text.

L266: The cutting area is not shown in Table 3, but in Table 4. Tables 3 and 4 were recreated to distinguish growth traits from carcass traits.

R. The comments were handled according to the suggestions of the examiner. As shown in the manuscript, tables 3 and 4 are renumbered.

L274-278: Clearly mark the table number.

R. The text clarified that the information of meat pH and color parameters corresponds to Table 5.

L274: It is called "higher pH", but what is higher? Is there any significant difference?.

R. The information requested by the examiner has been included in the text, and the information in Table 5 clarifies this question.

L275: The letter "H" appears in the list, but it is lowercase in Table 5. Correct with the same symbol.

R. Comments of reviewers have been discussed in Table 5.

L276: Quote the previous research and describe the normal range.

R. The discussion chapter contains the information pointed out by the examiner, with corresponding bibliographic references.

L298-304: Even compared with numbers, the explanation is difficult to understand.

R. The paragraph has been revised and corrected, and the wording is concise and clear, and the results are explained in a way similar to other publications on genes related to lipid metabolism.

Table 3: HZ ->ZH

R. The observations in Table 3 are discussed.

Table 4: CZ ->ZH

R. The observations in Table 4 are discussed.

Author Response

Comentarios para agricultura-2230437 -peer-review-v2

El manuscrito titulado “Efecto del clorhidrato de zilpaterol y metionina de zinc sobre el crecimiento, las características de la canal, la calidad de la carne, el perfil de ácidos grasos y la expresión génica en el músculo longissimus dorsi de ovejas en engorde intensivo” evaluó el efecto de ZH y ZM en la dieta sobre la eficiencia alimenticia. , características de la canal, calidad de la carne, contenido de ácidos grasos y expresión de genes relacionados con el metabolismo de los lípidos en el músculo longissimus dorsi de ovejas de engorde.

Este estudio tiene un cierto significado práctico para la producción ovina. Sin embargo, el nivel de redacción de este artículo debe mejorarse. El artículo necesita una amplia modificación antes de considerar su publicación.

R. Most of the comments and suggestions for format, writing, syntax and punctuation, as well as content, indicated by the reviewers, were included in the text of the manuscript.

The following are the main defects of this manuscript.

  1. There are many grammatical and formatting errors in this article. The expression of important research results and their meanings are unclear, which is easy to cause ambiguity. And some of the mistakes are caused by carelessness. Even some statements in this article are inconsistent. Please check the manuscript carefully to revise every grammatical/editorial errors. It is suggested to provide an English editing certificate.

R. The entire manuscript was reviewed for its wording, syntax, and punctuation as recommended by the reviewer, and the observations of each section, noted in the report, were addressed.

  1. Follow more accurately the Guide for Agriculture.

  1. R. The corresponding corrections were made according to the authors' guide of the Agriculture journal.

  1. In the discussion part, there is too much theoretical knowledge and too little content to compare the results of this experiment with those of other researchers. More discussion with other research results should be added.

  1. R. This observation was addressed in the manuscript, the discussion in each section was expanded by comparing the results of other investigations with those of the present study.

  1. Please provide the full names of the abbreviations when they appear in the summary or in the text for the first time. E.g., “LW” in Line 26, “DM” in Line 30, “IMF” in Line 54, “WB” in Line 55, “ZH” in Line 63, “FA” in Line 69, “ZM” in Line 84, “SFA and UFA” in Line 102, “DWG” in Line 135, and so on.

R. The comment and suggestions of the reviewer were addressed in the text, the full names of the abbreviations were included.

  1. In the introduction and discussion parts, there are few introductions and discussions about ZM. It should be supplemented.

R. Introductory information on the additive Zn methionine was included in the manuscript as requested by the reviewer.

Line 30: add “The results showed that” before “ZH increased….”

R. The reviewer's comment in the abstract section was addressed. Suggested words were added to the statement.

Line 31: ether extract in IMF is repeat. In fact, ether extract indicated the fat content in LD muscle in this test.

R. The reviewer's pertinent observation in the text of the abstract was addressed.

Line 41-43: According to Figure 1, ZM had no significant effect on the expression of DGAT1 gene (P > 0.05).

R. The observation was addressed in the text, indeed, as indicated in figure 1, there was no effect of ZM on the expression of the DGAT1 enzyme gene.

Line 84: What is ZM? There is no previous introduction for it. Zinc and ZM are definitely different. Please supplement the introduction of ZM, including its research status.

R. Information on the Zn methionine complex was included, as well as results of previous research on the effect of its inclusion in diets for cattle and sheep in feed lot systems.

Line 106-111: This paragraph is repeated and cumbersome.

R. The paragraph was reduced and its wording simplified.

Line 125-126: The dosage and method of ZH and ZM should be provided here.

R. The reviewer's observation was addressed, including the doses of each additive supplied.

Line 128: What is CZ and PV?

R. Corrected the corresponding abbreviations.

Line 144: change four hours to 4 h.

R. The reviewer's comment was addressed.

Line 207: put parentheses around ΣSFA+ΣPUFA in the two formulas.

R. The reviewer's comment was addressed.

Line 207: Is this formula “S/P=(C14:0+C16:0+C18:0)/ΣSFA+ΣPUFA” right, for S/P?

R. The reviewer's observation was addressed by placing the correct formula and its bibliographic support.

Line 231: the housekeeping gene is ribosomal protein S9 in Line 207 or GAPDH in Line 241? And provide the primers information of housekeeping gene in Table 2.

R. The information requested by the reviewer was included in the text of manuscript and in the Table 2.

Line 245: change results to data.

R. The reviewer's comment was addressed.

Line 251: What is (1)?

R. Antedependence: First order. This covariance structure has heterogeneous variances and heterogeneous correlations between adjacent items. The correlation between two non-adjacent elements is the product of the correlations between the elements that lie between the elements of interest.

Line 251: What is FI? It was not showed in Table 3.

R. Corrected the abbreviation as in table 3 (dry matter intake, DMI)

Line 251-253: What is the meaning of this sentence?

R. The reviewer's observation was addressed, the wording was corrected.

Line 261: where was FE and PCF in Table 3? Check the data carefully.

R. The observation in tables 3 and 4 was corrected as indicated by the reviewer (feed conversion and CCW).

Line 261: ≤ was wrong, it was =, according to Table 3.

R. The symbol ≤ has been corrected by =, as indicated in Table 3.

Line 266: chop area was not in Table 3, in fact in Table 4.

R. Corrected remark by redoing Table 4 as suggested by Reviewer 1.

Line 300-301: the genes of the enzymes, HSL, MGL and DGAT1 were expressed, respectively, 0.11, 0.69, 0.61 times more than the group without ZM. According to the original data, is this sentence, right?

R. The sentence is correct, the paragraph was redrafted contemplating the 4 enzymes affected by ZM.

Line 301-304: in fact, the P value of interaction ZM with ZH on MGL in Figure 1 was > 0.05!

R. The error was corrected due to the judicious observation of the reviewer, the interaction was not significant in the expression of this MGL gene.

Line 306-311: The whole paragraph is only one sentence? A sentence that can fully express a meaning is a complete sentence, which should be followed by a full stop, instead of using semicolons. There were many semicolons in this article.

R. The wording, syntax, and punctuation of this paragraph and the rest of the manuscript were reviewed and corrected.

Line 320: The number of decimal places in Table 3 is 1, 2, or even 3. The decimal places of data should be consistent in one table, also for all the tables.

R. The decimal values were homogenized to two or three depending on the variable and table.

Line 320: What is TGW in Table 3?

R. It means total gain weight, which is indicated at the bottom of the table.

Line 320: Kg or kg in Table 3?

R. It was corrected in table 3 (kg)

Line 322-323: Why is ZH based on LW, while ZM based on DM?

R. It was decided to do the calculations like this since they are the doses recommended by the manufacturers of both additives.

Line 322: Why was “LW” here, while “PV” below Table 4, 5, 6, and 7?

R. Fixed error on tables 4-7 placing LW.

Line 328: where is FE, feed efficiency in Table 3?

R. This variable was removed from the text, it was not included in Table 3.

Line 338: use mg/kg unified in the text instead of ppm.

R. The unit mg kg-1 was standardized throughout the manuscript instead of ppm.

Line 352: What is the meaning of CZ in Table 4?

R. CZ was corrected for ZH in Table 4.

Line 352: Where is the area of chop area? The LD muscle?

R. Table 4 was redone with variables of carcass characteristics/traits and meat quality.

Line 352: What is the meaning of “g 100 g”? is it g/100 g?

R. 100 g= g/100 g, corrected in table 4.

Line 358-359: Kidney fat scores 1-4 here, but it was showed 1-3 in Table 4.

R. The score was corrected in the table, the scale is from 1-4 points.

Line 364: What is de pH?

R. The final pH values (6.13 and 6.06) (24 h post sacrifice) were included in the text.

Line 413: it’s better to merge Table 6 and Table 7, since both of the two tables showed the fatty acids aspect.

R. Initially they were merged but the table was very large.

Line 422-423: there was two meanings for UFA here, and two meanings for PUFA. This is definitely not feasible. The meanings are not clear.

R. This observation was addressed, the duplication of values was eliminated, placing them in single line use.

Line 503: delete the full name of DGAT1, since it’s provided before.

R. The reviewer's observation was addressed, leaving only the abbreviation.

Line 507-508: ZH interaction with ZM did not affect the mRNA expression of the MGL gene, since it’s showed ZH*ZM P>0.05 in Figure 1.

R. The wording in the text of the manuscript was corrected.

Line 520-522: This sentence has nothing to do with this article.

R. The sentence and bibliographical reference of the manuscript were removed.

Line 524: in Figure 1, “ZINC” was showed, not ZM. In fact, ZINC and ZM are different.

R. This observation has been corrected in Figure 1.

Line 524: the DGTA1 in Figure 1 was wrong, it should be DGAT1. Also, in the footnote.

R. Corrected the abbreviation DGAT1 in figure 1.

Line 528-529: ZH: 0 or 0.2 mg kg-1 LW. ZM: 0 or 80 mg kg-1 DM.

R. The observation in the title of Figure 1 was addressed.

Line 532: According to the data provided, it cannot improve the growth rate, but only reduce the feed conversion.

R. Se abordó la observación, indicando en las conclusiones lo que favorecía el efecto únicamente en la conversión alimenticia.

Round 2

Reviewer 2 Report

I am satisfied with the revisions that have been made by the authors.